# An Unsupervised Deep-Transfer-Learning-Based Motor Imagery EEG Classification Scheme for Brain–Computer Interface

**DOI:** 10.3390/s22062241

**Published:** 2022-03-14

**Authors:** Xuying Wang, Rui Yang, Mengjie Huang

**Affiliations:** 1School of Advanced Technology, Xi’an Jiaotong-Liverpool University, Suzhou 215123, China; xuying.wang19@alumni.xjtlu.edu.cn (X.W.); r.yang@xjtlu.edu.cn (R.Y.); 2School of Electrical Engineering, Electronics & Computer Science, University of Liverpool, Liverpool L69 3BX, UK; 3Research Institute of Big Data Analytics, Xi’an Jiaotong-Liverpool University, Suzhou 215123, China; 4Design School, Xi’an Jiaotong-Liverpool University, Suzhou 215123, China

**Keywords:** brain–computer interface, motor imagery, electroencephalography, transfer learning, common spatial pattern

## Abstract

Brain–computer interface (BCI) research has attracted worldwide attention and has been rapidly developed. As one well-known non-invasive BCI technique, electroencephalography (EEG) records the brain’s electrical signals from the scalp surface area. However, due to the non-stationary nature of the EEG signal, the distribution of the data collected at different times or from different subjects may be different. These problems affect the performance of the BCI system and limit the scope of its practical application. In this study, an unsupervised deep-transfer-learning-based method was proposed to deal with the current limitations of BCI systems by applying the idea of transfer learning to the classification of motor imagery EEG signals. The Euclidean space data alignment (EA) approach was adopted to align the covariance matrix of source and target domain EEG data in Euclidean space. Then, the common spatial pattern (CSP) was used to extract features from the aligned data matrix, and the deep convolutional neural network (CNN) was applied for EEG classification. The effectiveness of the proposed method has been verified through the experiment results based on public EEG datasets by comparing with the other four methods.

## 1. Introduction

The brain is closely related to human physiological functions and psychological activities, so brain science research plays a significant role in promoting human health and well-being. In the past decades, human enthusiasm for brain research has been gradually increasing. The US government named the last decade of the twentieth century “the decade of the brain” [1], and the Fourth Neuroscience Conference held in Japan declared that the 21st century is the “era of brain science”. American scientist Vidal first proposed the prototype of brain–computer interface (BCI) technology in 1973 and he defined BCI technology as a computer technology that can reflect the brain’s internal micro-information [2]. The first International Brain–Computer Interface Conference, held in 1999, officially proposed the concept of BCI for the first time—BCI is a system that transmits information between the brain and computer or other external equipment. The original intention of BCI system research is to provide a new method for those who have normal thinking but suffer from physical dysfunction to communicate with the external environment. In recent years, the further development of BCI technology has brought hope to the people with disability who cannot take care of themselves. The successful application of BCI technology in medical rehabilitation can greatly improve the quality of patients’ lives. In addition to medical treatment and rehabilitation, BCI technology also has broader practical significance and various application fields, including the military field, transportation, and daily entertainment [3,4,5,6,7].

Since the German expert Hans Berger first discovered the electrical activity of the cerebral cortex and recorded it in the form of electroencephalograph (EEG) in 1929 [8], scientists started to collect EEG signals to study the connection between the brain and motor nervous system diseases and diagnose neurological diseases, such as gradual freezing, myasthenia gravis, epilepsy, and other neurological diseases. Kovari et al. made an extensive study of algorithmic problem-solving and executive function [9] and proposed a wide variety of applications of modern human–computer interfaces based on eye tracking [10], eye–hand coordination [11,12], and brain waves [13]. Costescu et al. investigated the use of GP3 eye tracker in assessing visual attention in children [14]. Maravic et al. utilized eye movement monitoring for the teaching process [15]. Motor imagery is a type of EEG signal defined as the mental execution of an action without any apparent movement or muscle activity [16]. The movement is only simulated repeatedly in the mind without obvious physical actions, but the brain’s motor-sensory cortex produces the same EEG signal that performs this movement at the same time. The research of motor imagery is generally to find a way to recognize whether the current brain performs the corresponding motion imagery.

A complete EEG-based BCI system consists of four parts: EEG signal acquisition, signal preprocessing, feature extraction, and pattern classification [17]. The signal acquisition is completed by the subject wearing an electrode cap to collect a specific area’s EEG signal. The preprocessing of the EEG signal includes filtering the signal in time, space, and frequency domains to reduce artifacts and noise interferences. The significant part, feature extraction, mainly refers to applying various time, frequency, or spatial domain processing techniques to extract the features that are conductive to classification, and reducing the dimension of feature vectors to meet the real-time system requirement. For pattern classification of EEG signal, the target is to design a suitable classifier for the extracted features to guarantee the system performance in accuracy.

For feature extraction methods, there are four commonly used algorithms: power spectrum density (PSD), sample entropy, wavelet transform, and common spatial pattern (CSP). PSD analyzes the signal from the perspective of frequency domain, and the purpose is to transform the EEG signal into a power spectrum and analyze the properties in the frequency domain such as power spectral density, energy, and amplitude. The PSA algorithm is simple and easy to operate, but cannot reflect the latent features. Sample entropy is a signal complexity analysis method and has the advantages of short analysis data, light computational load, and strong anti-interference [18]. As a nonlinear analysis algorithm, sample entropy can mine the nonlinear characteristics of EEG signals but cannot reflect time–frequency features of EEG signals. Wavelet transform is a typical time-spectrum analysis method to extract the local characteristics in specific time and frequency segments by using wavelet functions to approximate the original signal. Wavelet transform can obtain the hidden time–frequency features with multi-resolutions, but needs to extract the frequency band in advance based on prior knowledge. Due to the complex properties of EEG signals, it is not easy to obtain accurate prior knowledge, therefore limiting the application flexibility of wavelet transform. CSP is currently considered as the most effective EEG feature extraction method [19]. The principle of CSP is to design a filter to maximize the variance of one kind of signal and minimize the other kind to obtain optimal features for classification. CSP does not need to select key frequency bands in advance, but is susceptible to external noise, thus requiring a combination with other methods to overcome this limit [20].

The current motor imagery EEG-based BCI system mostly uses conventional signal processing or machine learning methods for feature extraction and classification [21,22,23]. The conventional machine learning methods usually follow two basic assumptions: (1) the training data and testing data should follow same statistical distribution; (2) there are enough labeled training samples to train the feature extraction or classification model. However, besides the common properties of EEG signal, such as nonlinearity and non-stationarity, motor imagery EEG signals have the following two main characteristics:

(1) Strong individual difference: The different motor imagery habits of individuals can lead to different motor imagery signals. Even for the same person performing the same motor imagery task at a different time, the incurred motor imagery signals will have some fluctuation and the interference from environment will intensify the difference. Thus, the abovementioned factors require the motor imagery signal processing algorithm to have strong adaptability and universality to ensure an accurate interpretation of the human consciousness and a successful human–computer interaction [24].

(2) Event-related synchronization and desynchronization: When persons make different limb motions, their active brain regions change, as do the potential frequencies generated. The amplitude of the beta rhythms (13~22Hz) and the mu rhythm (8~12Hz) of the sensorimotor cortex on the left side of the brain rises dramatically while envisioning the movement of the left limb, resulting in event-related synchronization (ERS); while the two rhythms of the right side sensorimotor cortex are significantly reduced, resulting in event-related desynchronization (ERD) [25,26]. According to this characteristic, the EEG data of the corresponding activated areas for specific motor imagery actions should be reserved for analysis, and the data with limited correlation should be discarded to reduce the computation complexity.

Based on the above analysis, it is often difficult and costly to obtain a large amount of labeled training data for the same subject in motor imagery EEG classification task. Moreover, due to the individual differences, it is infeasible to directly use the EEG data of other subjects to train the classification model for new target subjects [27]. Therefore, this paper aims to propose an unsupervised transfer learning scheme to build reliable and accurate feature extraction and classification model for motor imagery EEG classification in BCI.

The main contributions of this paper areas follows: (1) an unsupervised feature extraction approach based on Euclidean space data alignment (EA) and CSP are proposed; (2) a deep transfer learning scheme based on the unsupervised feature extraction and the deep convolutional neural network classifier is established for motor imagery EEG classification. The rest of this paper is organized as follows: Section 2 introduces the basic concepts and the proposed algorithm, Section 3 discusses the datasets used in this study and the experiment results, and Section 4 concludes this paper.

## 2. Methods

### 2.1. Wavelet Denoise-Based Preprocessing

The EEG signal collected from the scalp is usually in microvolt level; thus, the acquisition process is prone to electromagnetic interference in the environment and the subject’s own electronystagmogram (ENG) and electromyography (EMG) signals. Therefore, the signal–noise ratio (SNR) of EEG signals is low, and improving the SNR is a prerequisite to guarantee the effective and reliable usage of EEG signal.

In this study, the wavelet analysis method was adopted to denoise and reconstruct EEG signals. Wavelet analysis is based on Fourier analysis with better time–frequency window characteristics and is more suitable for the analysis of non-stationary signals such as EEG signals than Fourier analysis.

### 2.2. Euclidean Space Data Alignment Based Preprocessing

As a type of transfer learning method, EA can realize similar distribution between the source domain data and the target domain data after Euclidean space transformation [28]. EA is an unsupervised method which does not require any labeled data of the target domain, and the general procedures are explained below.

The signal matrix of $X_{i},X_{i},\in R^{M\times N}$ can be produced from the ith trial of motor imagery signals, where *M* is the signal channel number and *N* is the sampling number. The spatial distribution is encoded by the covariance matrix $\Sigma_{i}$ of $X_{i}$ of a motor imagery trial, which can be calculated from the equation below:(1)$$\Sigma_{i}=X_{i}X_{i}^{T}$$

Then, the following reference matrix $\bar{R}$ can be used in the EA method:(2)$$\bar{R}=\frac{1}{n}\sum_{i=1}^{n}X_{i}X_{i}^{T}$$
where *n* is the subject trial number.

Then, the alignment process is conducted based on the following equation:(3)$${\tilde{X}}_{i}={\bar{R}}^{-1/2}X_{i}$$

The mean covariance matrix of all *n* aligned trials is
(4)$$\begin{array}{cc} \frac{1}{n}\sum_{i=1}^{n}{\tilde{X}}_{i}{\tilde{X}}_{i}^{T} & =\frac{1}{n}\sum_{i=1}^{n}{\bar{R}}^{-1/2}X_{i}X_{i}^{T}{\bar{R}}^{-1/2}\\ \; & ={\bar{R}}^{-1/2}\left(\frac{1}{n}\sum_{i=1}^{n}X_{i}X_{i}^{T}\right){\bar{R}}^{-1/2}\\ \; & ={\bar{R}}^{-1/2}\bar{R}{\bar{R}}^{-1/2}=I \end{array}$$

From the above analysis, the mean covariance matrices will become the identity matrix after data alignment for all subjects. The same mean covariance matrices of all subjects indicate that the spatial distributions are becoming the same, which is an important prerequisite in transfer learning. Then, the aligned signal matrix can be used for further data processing such as feature extraction.

### 2.3. Common-Spatial-Pattern-Based Feature Extraction

CSP is a supervised two-class spatial filtering algorithm, which was first proposed by Koles et al. in 1990 [29]. The basic principle of CSP is to find a set of optimal spatial filters for projection and obtain features with high discrimination vector, by maximizing the variance of one type and minimizing the variance of the other type. The CSP algorithm to extract the features of the two-class motor imagery EEG signals is explained below.

Suppose the EEG signal of a certain motor imagery trial is expressed as an M×T dimensional matrix *X*, where *M* is the number of EEG channels and *T* is the number of sampling points for each channel. The average covariance of the two categories can be derived as follows:(5)$$C_{y}=\frac{1}{n_{y}}\sum_{i=1}^{n_{y}}\frac{X_{i}\left(y\right)X_{i}^{T}\left(y\right)}{\mathrm{trace}{\left(X_{i}\left(y\right)X_{i}^{T}\left(y\right)\right)}^{\prime }}$$
where y∈{1,2} denotes different class, $X_{i}$ represents the *i*th sample, $X_{i}^{T}$ is the transpose of $X_{i}$, trace(·) denotes the trace of the matrix, $n_{y}$ is the number of samples of class *y*, and $C_{y}$ is the average covariance of class *y*. Then, the covariance of the mixed space is obtained by superposing the average covariance of different classes:(6)$$C_{c}=\sum_{y}C_{y}$$

We perform eigenvalue decomposition on the mixed space covariance matrix:(7)$$C_{c}=U_{c}\lambda_{c}U_{c}^{T}$$
where $U_{c}$ is feature vector matrix and $\lambda_{c}$ is a diagonal array of corresponding eigenvalues. We sort the eigenvalues in descending order and compute the whitening matrix as follows:(8)$$P=\sqrt{\lambda_{c}^{-1}}U_{c}^{T}$$
where *P* is the whitening matrix for spatial decorrelation to transform the covariance matrix of Y=PX into a diagonal matrix. Then, we apply the whitening matrix to the average covariance matrix of the two classes:(9)$$S_{y}=PC_{y}P^{T}$$

The principal component decomposition of the matrix after whitening is
(10)$$S_{1}=B\lambda_{1}B^{T},S_{2}=B\lambda_{2}B^{T},\lambda_{1}+\lambda_{2}=I$$

After decomposition, $S_{1}$ and $S_{2}$ have the same eigenvector matrix *B*, and the sum of the eigenvalue matrices of two classes obtained by the decomposition is the identity matrix *I*. The eigenvector corresponding to the largest eigenvalue of $S_{1}$ minimizes the eigenvalue of $S_{2}$, and vice versa. Therefore, the difference between the two classes of signals is maximized. Using the whitening matrix *P* and the decomposed eigenvector matrix *B*, the projection matrix can be obtained as
(11)$$W=B^{T}P$$

The original signal $X_{i}$ after projection is
(12)$$Z_{i}=WX_{i}$$

Finally, the features can be obtained as follows:(13)$$x_{r}=\log\frac{\mathrm{var}\left(Z_{r}\right)}{\sum_{j=1}^{2m}\mathrm{var}\left(Z_{j}\right)}$$
where var(·) indicates variance, *r* represents the *r*th row of signal, 2m represents the first and last representative *m* lines, and $x_{r}$ is the eigenvalue computed from the *r*th row.

### 2.4. Convolutional-Neural-Network-Based Classification

Deep learning algorithms have developed rapidly in recent years, and their powerful feature extraction capabilities have made them favored by more and more researchers [30,31]. Through multiple hidden layers in feature learning, the multi-dimensional raw data can be transformed into low-dimensional encoded features. Therefore, the convolutional neural network (CNN) algorithm can always provide reliable and good performance compared with other networks. The CNN structure is mainly composed of convolution layer, downsampling layer, and fully connected layer [32]. The local features can be extracted from the input data through the convolution layer, and can be simplified via the downsampling layer. After alternately applying these two layers, the fully connected layer is used to generate the output as classification result.

The convolution layer is the crucial layer in CNN, and the specific operation is to perform convolution operations on different kernels with input data through forwarding operations. Therefore, different feature maps of the same input data can be obtained. The value of the feature map coordinates (i,j) with *k*th convolution kernel is shown below:(14)$$y_{i,j}=f\left(W{\left(l\right)}_{k}*I\left(i,j\right)+b\right)$$
where *f* is the activation function of the neuron, *b* is the bias, $W{\left(l\right)}_{k}$ is the *k*th convolution kernel of *l*th layer, and I(i,j) is the input data of convolution layer. For the activation function, rectified linear unit is used in this study:(15)h(x)=max(0,x)
where *x* is the input of the unit.

In order to reduce the dimension of the outputs and weights network and avoid overfitting, the pooling layer is adopted to downsample the obtained feature maps $x_{i,j}$. In this study, the max pooling algorithm is adopted. The computation of the pooling layer with pooling kernel size k×k is shown below:(16)$$g\left(x\right)=\max\left(x_{\left[i,i+k][j,j+k\right]}\right)$$

The fully connected layer is used to classify the extracted features after the last pooling layer. The output of the neuron in the fully connected layer is computed as follows:(17)$$y_{l}=h\left(w^{*}y_{l-1}+b\right)$$
where *f* is the activation function of the neuron, *l* and l−1 represent the current and the previous layers, and *b* is the bias.

## 3. Experiment and Result Analysis

In this study, a deep transfer learning scheme based on the unsupervised feature extraction and deep convolutional neural network classifier was proposed for motor imagery EEG classification. To illustrate the effectiveness of the proposed method, experiments were conducted based on motor imagery EEG datasets with two comparison approaches, support-vector-machine (SVM)-based approach and fine-tuning CNN approach. This section introduces the experiment paradigm and the experiment results with discussion. The flowchart of the proposed unsupervised deep transfer learning method and two comparison methods is shown in Figure 1:

### 3.1. Dataset Description

The dataset of “two class motor imagery (002-2014)” from BNCI Horizon 2020 was used in the experiments [33]. Figure 2 shows the electrode positions and the experiment paradigm for this dataset.

The cue-guided Graz-BCI training paradigm was used as the guideline in this study [34]. In this paradigm, the recording, training, and feedback were all conducted within the same session. There were eight runs in each session: five runs for training purpose, and the other three runs for validation purpose. There were 20 trials in each run. The participants performed the task of sustained motor imagery of right hand or of feet for 5 s as instructed by the cue. Each trial lasted for 10 s: the first 2 s for resting; at 2 s, a beep signal was issued as preparation reminder; at 3 s, the arrow showed the motor imagery cue on the screen and the subject started the motor imagery task according to the cue. The EEG signal was measured with Ag/AgCl electrodes at a sampling rate of 512 Hz. A total of 15 electrodes were used during the data collection, divided into three groups. Each group has one center electrode and four extra electrodes placed 2.5 cm from the center electrode. The three center electrodes were placed at positions C3, Cz, and C4, as shown in Figure 2. The reference and ground electrodes were mounted on the left and right mastoid positions, respectively. The 14 participants were aged between 20 and 30 years, with no medical or neurological diseases.

To obtain the original signal matrix, the sampling points were selected from 0.5 s after the cue of each trial with time duration of 4 s. For each subject, the signal matrix of 15×2048×100 (channels × sampling points × trials) was generated for further processing. To divide the source domain and target domain, five training runs were used in this study. The combined signals of the first ten subjects were used as the source domain, and the combined signals of the last four subjects were used as the target domain.

### 3.2. Denoise Process

The signal denoise for both source and target domain signals were conducted using wavelet decomposition algorithm. Taking the C3 electrode located in the left hemisphere motor area as an example, the respective electrode position is shown in Figure 2. The original motor imagery EEG signals collected for two classes of right-hand and feet are shown in Figure 3.

In this study, the Daubechies wavelet db6 function was adopted to decompose the collected motor imagery EEG signals into four frequency bands: 0–6.5 Hz, 6.5–12.5 Hz, 12.5–25 Hz, and 25–50 Hz. Then, only the two main frequency bands 6.5–12.5 Hz and 12.5–25 Hz were retained during the EEG signal reconstruction. The EEG signal preprocessed by wavelet analysis can reduce noise and enhance features for subsequent feature extraction and classification.

To illustrate the effect of of denoise and enhancement of the original signal, we take the C3 original signal when performing right-hand MI as an example to show the process of wavelet denoise. Figure 4a,b show the time-domain analysis of the original signal and reconstructed signal.

### 3.3. Data Alignment

The EA algorithm was adopted to realize the data alignment of the source domain and target domain. To illustrate the effect of data alignment, we performed t-stochastic neighbor embedding (t-SNE) [35] on the source and target domains CSP matrix before and after alignment. The t-SNE visualization is shown in Figure 5. The blue and green dots represent the t-SNE features of two different MI scenarios, feet and right-hand, respectively. Specifically, the red dots correspond to the t-SNE visualization of the two MI tasks in the target domain. It can be demonstrated that the data of each domain are scattered in the feature space before EA. Then, after alignment, the features of both domains are overlapped with each other. Thus, the classifier trained on the source domain data can be transferred to the target domain for utilization.

### 3.4. Feature Extraction

The CSP algorithm is used in feature extraction for both aligned source domain and target domain signal matrix. For each 15×2048 aligned matrix, we set 13 pairs of spatial filters for projection and feature extraction, then the feature vector with size of 100×1 can be obtained, which can be used for subsequent classification, as shown in Figure 6. The extracted features are used for subsequent classification.

### 3.5. Pattern Classification

For each 100×1 feature matrix, the first step was to reshape it into a 2D matrix in shape of 10×10. Then, the reshaped feature matrix was used as the input of 2D CNN, with two convolutional layers and two maxpooling layers. Then, a normal fully connected layer can be used to perform classification. For each subject in the target domain, 80% of the trials are randomly selected as the training set, and the remaining 20% are selected as the test set. In the training process of the proposed method ES-CSP-CNN, the learning rate of CNN is set to 0.4, ReLU is set as an activation function, and maxpooling is adopted as the downsampling method. The training epoch and batch size are set as 160 and 1. The respective feature reshape and CNN structure are shown in Figure 7. The training process and loss of CNN are shown in Figure 8.

### 3.6. Comparison Methods

To illustrate the effectiveness of the proposed method, two comparison approaches were used in this study to compare the classification performances as shown in Figure 1. The first one is SVM-based approach, using the feature matrix of the source domain after CSP to train the SVM classifier. The second one is fine-tuning-CNN-based approach. This fine-tuning CNN architecture is composed of six convolutional layers and five pooling layers along with one softmax layer to classify motor imagery tasks, as shown in Table 1. The source domain signal matrices were used as input for the training of CNN, then the weights of the top five layers were frozen and the classifier was retrained using one subject’s EEG signal matrices in the target domain. The flowchart of the fine-tuning CNN is shown in Figure 9.

### 3.7. Results Discussion

The overall classification accuracy of the proposed method and the comparison methods on four target subjects are shown in Table 2, where EA-CSP-CNN stands for the proposed method, EA-CSP-SVM stands for the SVM-based approach with EA and CSP, and EA-ftCNN stands for the fine-tuning-CNN-based approach with EA. It can be seen that the proposed method (EA-CSP-CNN) can achieve the high accuracy in all four target subjects. For individual comparisons, the improvements of EA-CSP-CNN are between 7% to 15% compared with EA-CSP-SVM, indicating that the secondary deep feature extraction mechanism further improves the feature representativeness. Moreover, the superior ability of CNN in nonlinear approximation compared to SVM also benefits the performance when fitting to EEG-CSP features. The improvements of EA-CSP-CNN are between 4% to 13% compared with EA-ftCNN. This result demonstrates the effectiveness of using CNN for the secondary deep feature extraction of EEG-CSP features, compared to using deep neural network with pretrain-finetune on the original EEG signal. More significantly, the proposed method does not rely on the source domain labels, which is more resource-efficient than EA-ftCNN.

## 4. Conclusions

Due to the strong individual difference and the limited correlation in motor imagery EEG signals, an unsupervised deep transfer learning scheme for motor imagery EEG classification was proposed in this paper to achieve reliable and accurate feature extraction and classification and reduce the computation complexity. The deep transfer learning scheme adopts EA and CSP to realize unsupervised feature extraction, and deep CNN is utilized as the classifier. The experiment results based on motor imagery EEG dataset illustrated the effectiveness of the proposed method by comparing with transfer-learning-based SVM method and fine-tuning CNN method. The future work can be conducted on improving the transfer learning algorithm by revising the data alignment method. In the conventional EA algorithm, the alignment of the Euclidean space only considers the alignment of the covariance matrix, but not the alignment of the overall probability distribution. Therefore, the probability distribution adaptation can be added to the alignment of the covariance matrix in the subsequent work.

## Figures and Tables

**Figure 1 sensors-22-02241-f001:**
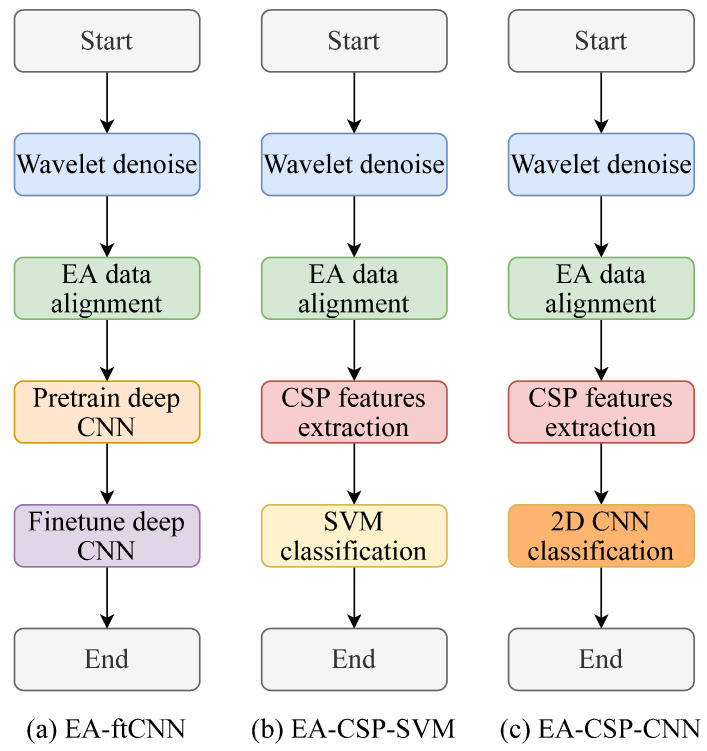
The experiment flowchart.

**Figure 2 sensors-22-02241-f002:**
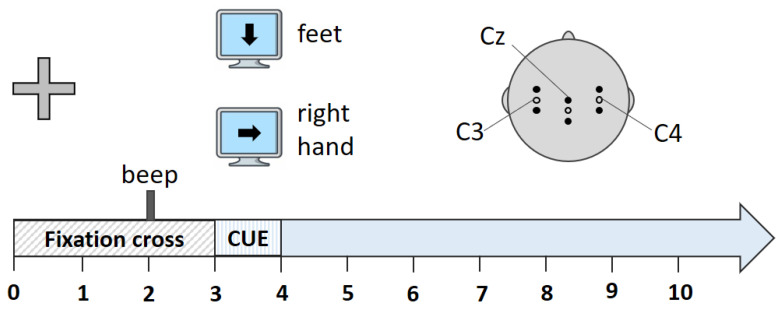
Experiment paradigm.

**Figure 3 sensors-22-02241-f003:**
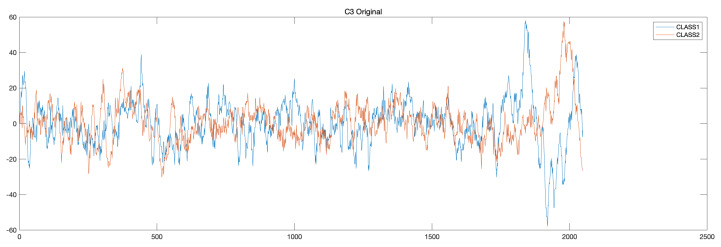
Motor imagery EEG signal of C3 of two classes.

**Figure 4 sensors-22-02241-f004:**
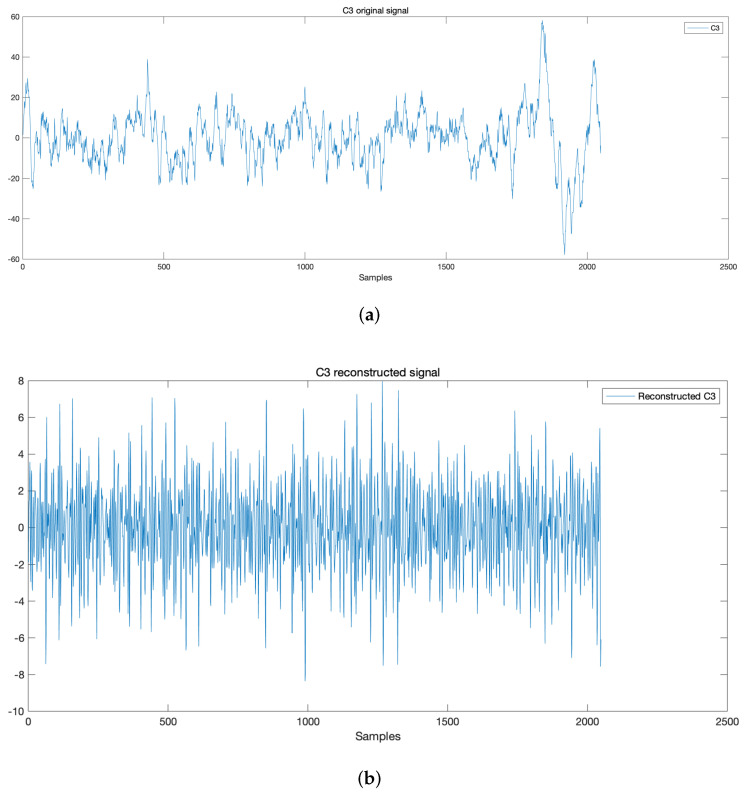
Three layers wavelet decomposition of C3. (**a**) Original signal; (**b**) Reconstructed signal.

**Figure 5 sensors-22-02241-f005:**
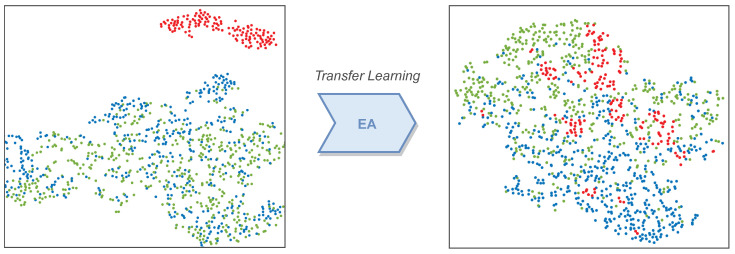
t-SNE visualization before (**left**) and after (**right**) EA.

**Figure 6 sensors-22-02241-f006:**
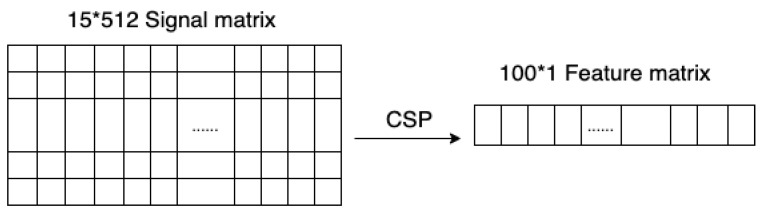
CSP feature extraction.

**Figure 7 sensors-22-02241-f007:**
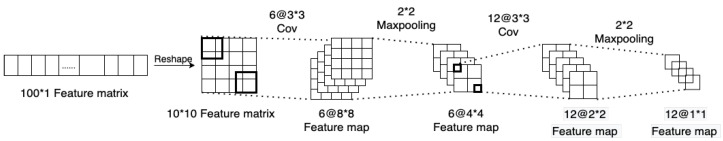
The structure of CNN.

**Figure 8 sensors-22-02241-f008:**
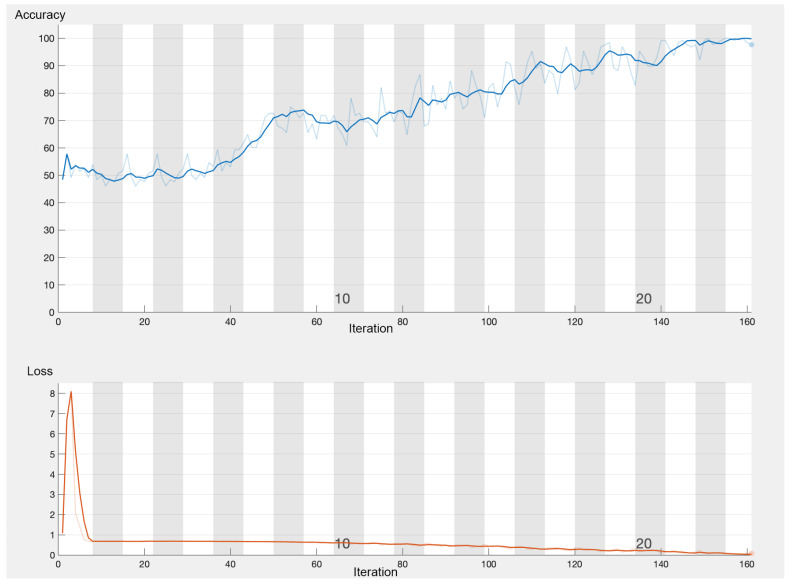
The training process and loss of CNN.

**Figure 9 sensors-22-02241-f009:**
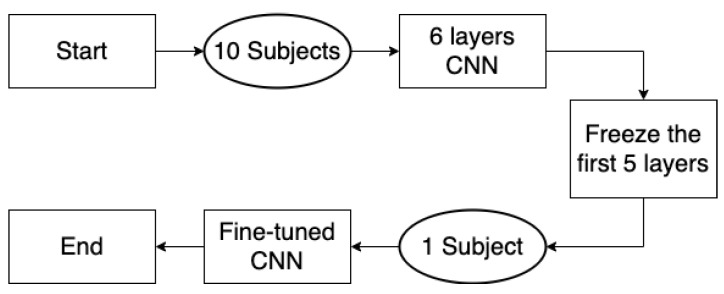
The flowchart of fine-tuning CNN.

**Table 1 sensors-22-02241-t001:** Details of the network structure.

No	Layer	Options
0	Input EEG	size = (250,250,1)
1	Convolutional layer	size = (250,250,1), kernel size = (11,11,32), padding = (1,1)
2	Maxpooling layer	size = (120,120,32), kernel size = (2,2,32), padding = (2,2)
3	Convolutional layer	size = (110,100,32), kernel size = (11,11,32), padding = (1,1)
4	Convolutional layer	size = (100,100,32), kernel size = (11,11,32), padding = (1,1)
5	Maxpooling layer	size = (50,50,32), kernel size = (2,2,32), padding = (2,2)
6	Convolutional layer	size = (44,44,64), kernel size = (7,7,64), padding = (1,1)
7	Maxpooling layer	size = (22,22,32), kernel size = (2,2,64), padding = (2,2)
8	Convolutional layer	size = (20,20,128), kernel size = (3,3,128), padding = (1,1)
9	Maxpooling layer	size = (10,10,128), kernel size = (2,2,128), padding = (2,2)
10	Convolutional layer	size = (8,8,128), kernel size = (3,3,128), padding = (1,1)
11	Maxpooling layer	size = (4,4,128), kernel size = (2,2,128), padding = (2,2)
12	Fully-Connected layer	size = (2048,1)
13	Softmax layer	size = 2

**Table 2 sensors-22-02241-t002:** Overall classification accuracy (%).

Target Subject	EA-CSP-SVM	EA-ftCNN	EA-CSP-CNN
S11	69	73	79
S12	72	64	87
S13	74	70	84
S14	60	63	67

## Data Availability

Restrictions apply to the availability of these data. Data was obtained from the open access BCI Dataset of BCI-Horizon 2020, and are available from http://www.bnci-horizon-2020.eu/database/data-sets (accessed on 1 February 2022) with the permission.

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
