# Peer review of "An Unsupervised Deep-Transfer-Learning-Based Motor Imagery EEG Classification Scheme for Brain–Computer Interface"

_sensors, 2022, doi:10.3390/s22062241_

Round 1

Reviewer 1 Report

This paper proposed a deep CNN model for motor imagery EEG classification enhanced with a feature extraction approach using Euclidean space data alignment and common spatial pattern (CSP). The idea is well presented and it is a worthy-research problem. However, there are inadequacies mentioned in the following:

  1. In Figure 1, it is better to separate between the flowchart for the proposed method and for the two comparison methods to avoid confusion.
  2. In Figure 5, authors should define the two different dot colors.
  3. The data explanation for training and validation has been well explained in the dataset description. Nonetheless, I think the data used for training and finetuning part should be different. In Figure 9, does the 1 subject used for finetuning belongs to the subject in the source domain as well?
  4. Transfer learning can also be done directly from the SVM. This could be another direction for the work.
  5. The discussion of the result is weak, please refer to other related works which are relevant to the proposed methods for comparison. The effect of EA-based preprocessing towards the result should also be discussed in detail.
  6. Some spelling/grammatical errors need to be fixed.

Reviewer 2 Report

The authors presented an unsupervised deep transfer learning-based method for BCI systems applying the idea of transfer learning to the classifi cation of motor imagery EEG signals. They used the Euclidean space data alignment methode to align the covariance matrix of source. Then they used the common spatial pattern methode to extract features from the aligned data matrix. They apply a deep convolutional neural network (CNN) for EEG classifi cation. They test the effectiveness of the proposed method based on public EEG datasets.

In the Introduction I would be glad to see a short paragraph introducing the wide variety of applications of modern human-computer interfaces. I would be happy to see some up-to-date interesting papers in the field, such as Examine the Effect of Different Web-based Media on Human BrainWaves, Study of Algorithmic Problem-Solving and Executive Function, Assessing Visual Attention in Children Using GP3 Eye Tracker, Quantitative Analysis of Relationship Between Visual Attention and Eye-Hand Coordination, Evaluation of Eye-Movement Metrics in a Software Debugging Task using GP3 Eye Tracker.

In the discussion part, the concrete results should be evaluated in more detail and more precisely, referring back to the results part. because the final result is only visible (accuracies) without detailed analysis.

How were you convinced of the validity and reliability of the system? Please describe the method of the evaluation considering the validity and reliability of the system.

Maybe much more detailed data analysis is requred to state correct conclusions supported by more precise statistical methods.

Please mention in the introduction that human-computer interface based mathods are also used in human-computer interface applications like Examination of Gaze Fixations Recorded during the Trail Making Test, Application possibilities of adaptive testing in information processing, Application of Eye Movement Monitoring Technique in Teaching Process, Examination of Gaze Fixations Recorded during the Trail Making Test.

Reviewer 3 Report

The paper presents an unsupervised deep transfer learning-based method to deal with the classification of motor imagery EEG signals.

The topic is very interesting. However, the content is not so innovative, because the proposed approach is different from the previous approaches proposed by the same authors only for the application of a well-known feature extraction techniques such as CSP. Moreover, I have several notes to do related to the experimental session:

1) why the authors select only two techniques for the comparison? And, why they select the compared approaches such as SVM and 3-layers CNN? To be honest, a description of the state of the art approaches for addressing the same problem should be introduced in order to motivate the performed comparison;

2) the whole experimental set-up is not clear. What is the split percentage between train and test data? What is the configuration of the hyper-parameters for the compared approaches? For example, what is the regularization value C for the SVM? Moreover, the used configuration for hyper-parameters is the optimal one for all compared approaches? Why the tuning procedure is performed only for the proposed approach? 

3) the discussion of the results is poor. Why the authors use only the accuracy? It could be interesting also to evaluate other measures such as f1-score. 

Round 2

Reviewer 1 Report

The authors' response is adequate and clearly explained. Minor issues should be fixed, such as:

  1. Line 127: usually in microvolt leve...
  2. Line 241: the proposed method ES-CSP-CNN...
  3. Line 255: The flowchart fo the finetuing...
  4. Eq. (15) and (16): please define what is 'x'

Reviewer 2 Report

I accept the paper with the additions.

Author Response

Thank you very much for your comment. We greatly appreciate your effort and time in reviewing our work.